# Parkinson's disease-associated ATP13A2/PARK9 functions as a lysosomal H+,K+-ATPase

Takuto Fujii [1] ✉, Shushi Nagamori [2], Pattama Wiriyasermkul [2], Shizhou Zheng[1], Asaka Yago[1], Takahiro Shimizu[1], Yoshiaki Tabuchi [3], Tomoyuki Okumura[4], Tsutomu Fujii [4], Hiroshi Takeshima [5] & Hideki Sakai [1] ✉

Mutations in the human ATP13A2 (PARK9), a lysosomal ATPase, cause Kufor-Rakeb Syndrome, an early-onset form of Parkinson's disease (PD). Here, we demonstrate that ATP13A2 functions as a lysosomal H+,K+-ATPase. The K+-dependent ATPase activity and the lysosomal K+-transport activity of ATP13A2 are inhibited by an inhibitor of sarco/endoplasmic reticulum Ca2+-ATPase, thapsigargin, and K+-competitive inhibitors of gastric H+,K+-ATPase, such as vonoprazan and SCH28080. Interestingly, these H+,K+-ATPase inhibitors cause lysosomal alkalinization and α-synuclein accumulation, which are pathological hallmarks of PD. Furthermore, PD-associated mutants of ATP13A2 show abnormal expression and function. Our results suggest that the H+/K+-transporting function of ATP13A2 contributes to acidification and α-synuclein degradation in lysosomes.

Parkinson's disease (PD) is one of the most common movement disorders. The neuropathological characteristic of PD is a progressive loss of dopaminergic neurons in the substantia nigra. Several disease-causing genes such as *α-Synuclein* (*PARK1*), *Parkin* (*PARK2*), and *ATP13A2* (*PARK9*) were found in PD[1,2]. Mutations in *ATP13A2* cause Kufor-Rakeb syndrome (Parkinson's disease 9), an autosomal recessive form of Parkinsonism with dementia[3–5]. In the human brain, ATP13A2 is significantly expressed in dopaminergic neurons of the substantia nigra[6]. Its localization was found in lysosomes of the neuronal cells[6,7]. ATP13A2 is a P5-ATPase belonging to the P-type ATPase family that undergoes autophosphorylation to couple ATP hydrolysis to cation transport during the catalytic cycle[8]. PD-associated mutations and knockdown of ATP13A2 cause lysosomal dysfunction and α-synuclein aggregation, which are pathological hallmarks of PD[6,7,9,10].

Polyamines, such as spermidine and spermine, are polycationic and aliphatic molecules participating in numerous cellular processes including gene transcription and translation, protein synthesis, cellular proliferation, and differentiation[11]. Recent reports suggest that ATP13A2 functions as a polyamine exporter in lysosomes[12,13]. Dysfunction of ATP13A2 prevents the polyamine export from the lysosomal lumen, leading to lysosomal polyamine accumulation and intraluminal alkalinization of lysosomes[12,13]. Abnormal cytoplasmic polyamine level promotes the aggregation and the fibrillization of α-synuclein[14]. ATP13A2 plays a protective role against the aggregation of α-synuclein by maintaining the integrity of the lysosomal membrane and also promotes the ATPase-independent secretion of α-synuclein through nanovesicles[15].

Most recently, the cryoelectron microscopy structures (cryo-EM) of polyamine-binding ATP13A2 were clarified[16–19]. Based on the structure, ATP13A2 may transport H+ into the lysosomal lumen[17]. On the other hand, ATP13A2 does not transport Mn2+ and Zn2+ [20], although it contributes to reducing the toxicity of these ions[21–24]. In rat neurons, modulation of ATP13A2 expression influences intracellular H+ and Ca2+ concentrations[6]. In human dopaminergic neurons, ATP13A2 deficiency changes cytosolic and lysosomal Ca2+ concentrations[25].

[1]Department of Pharmacological Physiology, Faculty of Pharmaceutical Sciences, University of Toyama, Toyama 930-0194, Japan. [2]Center for SI Medical Research and Department of Laboratory Medicine, The Jikei University School of Medicine, Tokyo 105-8461, Japan. [3]Division of Molecular Genetics Research, Life Science Research Center, University of Toyama, Toyama 930-0194, Japan. [4]Department of Surgery and Science, Faculty of Medicine, University of Toyama, Toyama 930-0194, Japan. [5]Department of Biological Chemistry, Graduate School of Pharmaceutical Sciences, Kyoto University, Kyoto 606-8501, Japan. ✉e-mail: fujiitk@pha.u-toyama.ac.jp; sakaih@pha.u-toyama.ac.jp

In the present study, we show that ATP13A2 functions as an $H^+,K^+$-ATPase in lysosomes. Interestingly, the $H^+/K^+$-transporting function of ATP13A2 is blocked by gastric proton pump ($H^+,K^+$-ATPase) inhibitors. These inhibitors induce lysosomal alkalinization and α-synuclein accumulation.

## Results

### Measurement of the ATP13A2-derived ATPase activity by using thapsigargin

First, we prepared the heterologous expression system of human ATP13A2 in HEK293 cells (Fig. 1a, inset). In the cells, ATP13A2 largely overlapped with a lysosomal marker LAMP2 (colocalization coefficient; $0.73 \pm 0.03$, $n = 12$) but not an endoplasmic reticulum marker calnexin (colocalization coefficient; $0.13 \pm 0.05$, $n = 14$) (Supplementary Fig. 1).

The ATP-hydrolyzing (ATPase) activity was measured in the membrane fractions of the cells at pH 6.5. Interestingly, ATP13A2 activity was sensitive to thapsigargin (TG), an inhibitor of sarco/endoplasmic reticulum $Ca^{2+}$-ATPases (SERCAs). As shown in Fig. 1a, low concentrations of TG (<100 nM) inhibited endogenous ATPase activity including $Ca^{2+}$-ATPases in both ATP13A2- and empty-vector transfected (mock) cells. In contrast, a high concentration of TG (5 μM)-sensitive ATPase activity in ATP13A2-transfected cells was much higher than in mock cells (Fig. 1a). Thus, the difference between 50 nM TG- and 5 μM TG-sensitive activities (ΔTG-sensitive ATPase activity) includes ATP13A2 activity (Fig. 1b). The ATP13A2-derived ATPase activity was disappeared in the mutant (D513N) in which the consensus motif for generating ATPase activity ($^{513}$DKTGT) was mutated (Fig. 1b). The expression level of D513N mutant was comparable to that of the wild-type (WT) ($96.2 \pm 9.6\%$ of

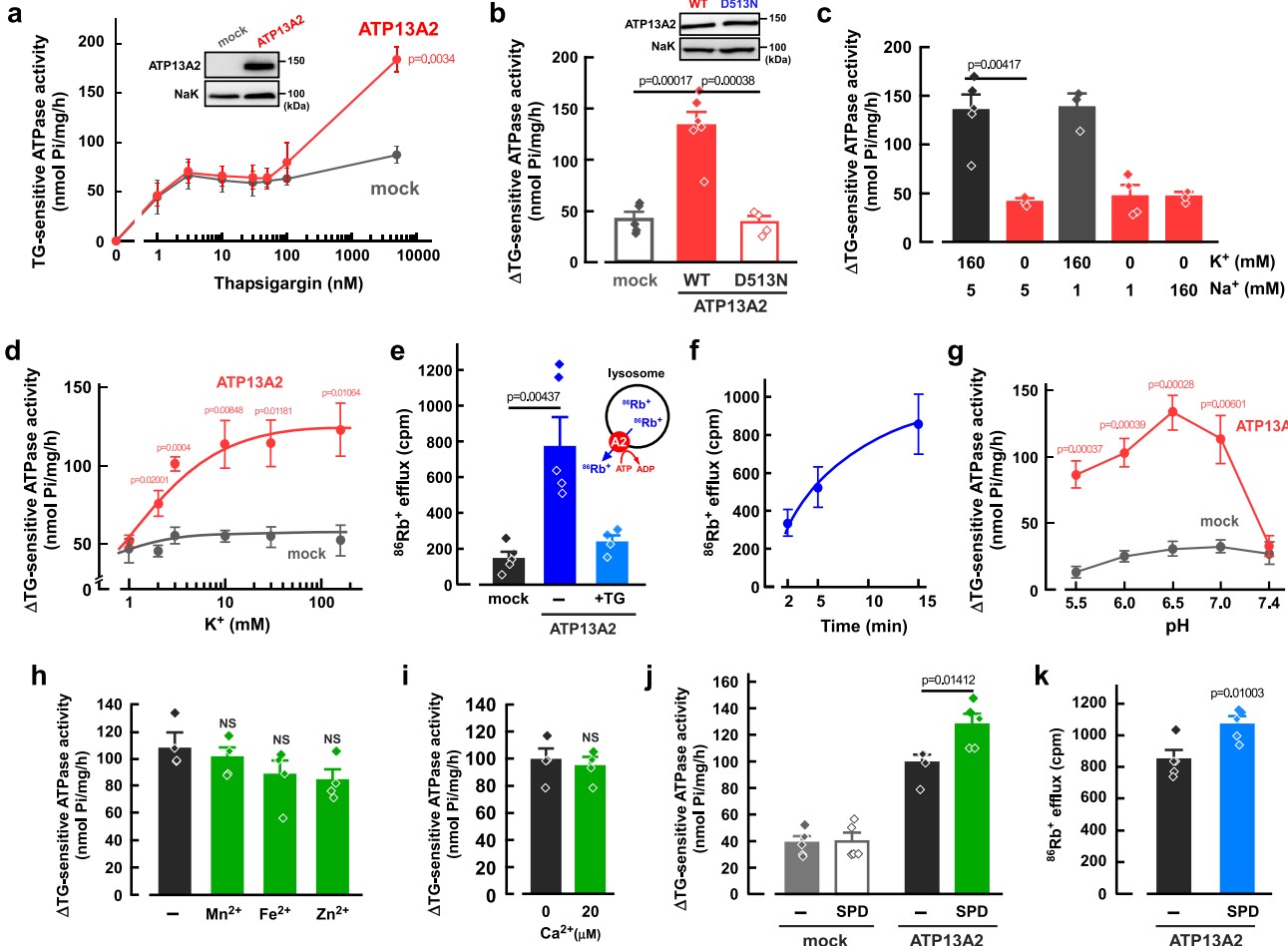

**Fig. 1 | Properties of cation transport of ATP13A2. a** Thapsigargin (TG)-sensitive ATPase activity in the ATP13A2- and mock-transfected HEK293 cells. ($n = 4$ independent replicates). Insert shows Western blot images of ATP13A2 (~150 kDa) and $Na^+,K^+$-ATPase (NaK; ~100 kDa). Statistical significance was determined by two-tailed unpaired Student's $t$ test. **b** ΔTG-sensitive ATPase activities (the difference between 50 nM TG- and 5 μM TG-sensitive activities) in the ATP13A2 WT-, D513N- and mock-transfected cells. ($n = 4$–$6$ independent replicates). Insert shows Western blot images of ATP13A2 (~150 kDa) and $Na^+,K^+$-ATPase (NaK; ~100 kDa). Statistical significance was determined by two-tailed unpaired Student's $t$ test. **c** ΔTG-sensitive ATPase activities in the ATP13A2-expressing HEK293 cells at various concentrations of $Na^+$ and $K^+$. ($n = 3$–$5$ independent replicates). Statistical significance was determined by two-tailed unpaired Student's $t$ test. **d** $K^+$-dependent activation of ΔTG-sensitive activity in the ATP13A2- and mock-transfected cells. ($n = 4$ independent replicates). Statistical significance was determined by one-way ANOVA. **e** ATP-dependent $^{86}Rb^+$ efflux from lysosomes of the ATP13A2- and mock-transfected HEK293 cells. Effect of TG (5 μM) on the $^{86}Rb^+$ efflux was examined. ($n = 4$–$5$

independent replicates). Statistical significance was determined by two-tailed unpaired Student's $t$ test. **f** Time-dependence of the $^{86}Rb^+$ efflux in the ATP13A2-transfected HEK293 cells. ($n = 4$ independent replicates). **g** pH-dependence of ΔTG-sensitive activity in the ATP13A2- and mock-transfected cells. ($n = 4$ independent replicates). Statistical significance was determined by two-tailed unpaired Student's $t$ test. **h, i** Effects of $Mn^{2+}$ (50 μM), $Fe^{2+}$ (50 μM), $Zn^{2+}$ (50 μM) (**h**), and $Ca^{2+}$ (0 and 20 μM) (**i**) on ΔTG-sensitive ATPase activity in the ATP13A2-transfected cells. ($n = 4$ independent replicates). Statistical significance was determined by two-tailed unpaired Student's $t$ test. NS, $P > 0.05$. **j** Effect of spermidine (SPD; 1 mM) on ΔTG-sensitive ATPase activity in the ATP13A2- and mock-transfected HEK293 cells. ($n = 5$ independent replicates). Statistical significance was determined by two-tailed unpaired Student's $t$ test. **k** Effect of spermidine (SPD; 1 mM) on the ATP-dependent $^{86}Rb^+$ efflux in lysosomes of the ATP13A2-transfected HEK293 cells. ($n = 4$-6 independent replicates). Statistical significance was determined by two-tailed unpaired Student's $t$ test. All data are presented as mean ± SEM. Source data are provided as a source data file.

WT, $n = 3$). We also established ATP13A2-knockout HEK293 (KO) cells by using the CRISPR-Cas9 genome-editing system. The faint expression of endogenous ATP13A2 disappeared in the KO cells (Supplementary Fig. 2a). In the D513N-transfected KO cells, no ATP13A2-derived ATPase activity was detected, while the WT-transfected KO cells showed significant activity (Supplementary Fig. 2b).

## Cation-transporting properties of ATP13A2

Next, the effects of $K^+$ and $Na^+$ on the ATP13A2-derived ATPase activity were examined (Fig. 1c). When $K^+$ was replaced with N-methyl-D-glucamine ($NMDG^+$) in the solution (0 mM $K^+$, 5 mM $Na^+$), the ATPase activity was dramatically reduced (Fig. 1c). In contrast, changes in $Na^+$ concentration had no significant effect on the activity (Fig. 1c). The ATP13A2-derived ATPase activity was increased in a concentration-dependent manner for $K^+$ with an $EC_{50}$ value of 2 mM (Fig. 1d). Since the $K^+$ concentration in the lysosomal lumen is much lower than that in the cytoplasm[26], we speculated that ATP13A2 exports $K^+$ from the lysosomal lumen to the cytoplasm. To evaluate $K^+$ transport by ATP13A2, the ATP13A2-transfected cells were cultured in the medium containing $^{86}Rb^+$ (a tracer for $K^+$). Then, the cells were treated with saponin (20 μg/ml) to permeabilize the plasma membrane, and ATP-dependent $^{86}Rb^+$ efflux was measured. Ouabain (30 μM), an inhibitor of $Na^+,K^+$-ATPase, had no significant effect on the ATP13A2-derived ATPase activity (Supplementary Fig. 3a) and the $^{86}Rb^+$ efflux (Supplementary Fig. 3b) in the ATP13A2-transfected cells, suggesting that the plasma membrane of the cells was completely permeabilized. The $^{86}Rb^+$ efflux in ATP13A2-transfected cells was significantly higher than that in mock cells and the efflux was inhibited by TG (Fig. 1e). The $^{86}Rb^+$ efflux in ATP13A2-transfected cells was increased in a time-dependent manner (Fig. 1f). These results suggest that ATP13A2 is a $K^+$-ATPase and exports $K^+$ from the lysosomal lumen to the cytoplasm.

The ATPase activity in ATP13A2-transfected cells exhibited a unique pH dependency; acidic pH increased the ATPase activity, and maximal activity was obtained at pH 6.5 (Fig. 1g). Interestingly, the ATP13A2-derived ATPase activity was not observed at pH 7.4 (Fig. 1g). Given that the $^{86}Rb^+$ efflux was detected under cytosolic solution at pH 7.4 (Fig. 1e, f), we speculated that the intraluminal acidic pH of the lysosomes facilitates $K^+$-transport of ATP13A2. On the other hand, $Mn^{2+}$, $Fe^{2+}$, $Zn^{2+}$, and $Ca^{2+}$, which relate to ATP13A2 function[6,21–25], had no significant effect on the activity (Fig. 1h, i). Besides, ATP13A2 exports polyamines (spermidine and spermine) from the lysosomal lumen to the cytoplasm[12]. Interestingly, spermidine (1 mM) significantly increased the ATPase activity in the ATP13A2-expressing cells but not in mock cells (Fig. 1j). In addition, the $^{86}Rb^+$ efflux from lysosomal lumen was significantly enhanced by the pretreatment of spermidine in the ATP13A2-expressing cells (Fig. 1k).

## Impairment of the ATP13A2 activity by PD-associated mutations

In PD patients, different types of point mutations of ATP13A2 have been found[3,4]. Among them, we generated four mutants (R449Q, G533R, A746T, and R980H), because these are located near the functional domains that are conserved in P-type ATPases[3,4,27] (Fig. 2a). In the HEK293 cells, expression levels of R449Q and A746T mutants were comparable to that of the WT, whereas those of G533R and R980H mutants were markedly reduced (Fig. 2b). The ATP13A2-derived activities of R449Q and A746T were significantly lower than the WT (Fig. 2c). Similarly, the ATP-dependent $^{86}Rb^+$ effluxes in the A746T- and R449Q-transfected cells were significantly smaller than the WT-transfected cells (Fig. 2d). Therefore, the loss of the ATP13A2 activity may be associated with the pathogenesis of PD.

## $H^+/K^+$-transporting activity of ATP13A2 is blocked by P-CABs

The gastric proton pump ($H^+$-$K^+$-ATPase) is responsible for acid secretion in the parietal cells of the stomach, and the pump is an essential therapeutic target for acid-related diseases[28]. Potassium-competitive acid blockers (P-CABs), such as SCH28080 and vonoprazan (TAK-438), selectively inhibit gastric $H^+$-$K^+$-ATPase in a $K^+$-competitive manner[29]. Vonoprazan is clinically used for the treatment of gastric ulcers and reflux esophagitis[29]. Interestingly, we found that the ATP13A2-derived ATPase activity is inhibited by SCH28080 and vonoprazan with $IC_{50}$ values of 1.2 and 0.8 μM, respectively (Fig. 3a, b). In addition, the ATPase activity was inhibited by bafilomycin A1, an inhibitor of vacuolar $H^+$-ATPase, with an $IC_{50}$ value of 0.5 nM (Fig. 3c). Another type of gastric $H^+$-$K^+$-ATPase inhibitors (proton pump inhibitors; PPIs), such as omeprazole, lansoprazole, and rabeprazole, are also clinically used drugs that covalently bind to cysteine residues at the luminal side of the ATPase[30]. Unlike P-CABs, neither omeprazole nor acid-treated omeprazole[31] (30 μM) inhibited the ATP13A2-derived ATPase activity (Supplementary Fig. 3c). Valinomycin, a $K^+$-selective ionophore, activates gastric $H^+,K^+$-ATPase activity in tubulovesicles[32]. However, it had no effect on the ATP13A2-derived ATPase activity (Supplementary Fig. 3c). We then examined the effects of SCH28080, vonoprazan, and bafilomycin A1 on the ATP13A2-derived $K^+$ efflux from the lysosomal lumen. In Fig. 3d, these drugs dramatically inhibited the ATP-dependent $^{86}Rb^+$ efflux in the ATP13A2-transfected cells. Given that the ATP13A2 activity was blocked by the $H^+,K^+$-ATPase inhibitors and the $H^+$-ATPase inhibitor and that the activity was increased by intraluminal acidic pH of the lysosomes (Fig. 1g), $H^+$ could be a substrate for ATP13A2.

To examine whether $H^+$ is transported by ATP13A2 in lysosomes, the intraluminal pH of the lysosomes was analyzed with FITC-dextran which shows a pH-dependent fluorescence. Because bafilomycin A1 inhibits vacuolar $H^+$-ATPase in addition to ATP13A2, significant increases in lysosomal pH were observed in both ATP13A2-transfected and mock cells (Fig. 3e). In contrast, SCH28080 and vonoprazan significantly increased the lysosomal pH in the ATP13A2-transfected cells but not in mock cells (Fig. 3e). These results suggest that ATP13A2 actually transports $H^+$ into the lysosomal lumen from the cytoplasm.

In the proposed catalytic cycle of the gastric $H^+,K^+$-ATPase, the active exchange of $H^+$ and $K^+$ is driven by ATP hydrolysis with cyclical conformational changes between two principle reaction states (E1 and E2) and their corresponding phosphoenzyme intermediates (E1P and E2P). Here, we measured the phosphorylation level of ATP13A2 using [γ-$^{32}$P]ATP. In the membrane fractions of ATP13A2-transfected cells, a phosphorylated band (~150 kDa) was detected at pH 6.5 in the absence of KCl (Supplementary Fig. 4a). This band was not observed in the mock-transfected cells (Supplementary Fig. 4a). The ATP13A2-derived phosphorylated band was significantly decreased by the addition of KCl (20 mM) (Supplementary Fig. 4a, b). In the absence of KCl, the phosphorylated level of ATP13A2 at pH 6.5 was higher than that at pH 7.4 (Supplementary Fig. 4c, d). These results are similar to the case for gastric $H^+,K^+$-ATPase[33], suggesting that ATP13A2 carries $H^+$ from the $(H^+)$E1P to the $(K^+)$E2P in its catalytic cycle.

## P-CABs induce lysosomal alkalinization and α-synuclein accumulation in neuroblastoma cells

We examined the properties of endogenous ATP13A2 activity in the human neuroblastoma SH-SY5Y cell line that is widely used in PD research as an in vitro model[34]. Knockout of ATP13A2 in the cells causes a reduction of polyamine transport and alkalinization of lysosomes[12]. Here, significant expression of ATP13A2 was observed in SH-SY5Y cells (Fig. 4a) and the expression was markedly decreased by transfection of siRNA for ATP13A2 (Fig. 4b). We then examined the $K^+$ and $H^+$ transport activity of endogenous ATP13A2 in lysosomes of the cells. TG-, SCH28080-, and vonoprazan-sensitive ATPase activities were observed in the membrane fractions of the control cells (Fig. 4b), which were significantly decreased by knockdown of ATP13A2

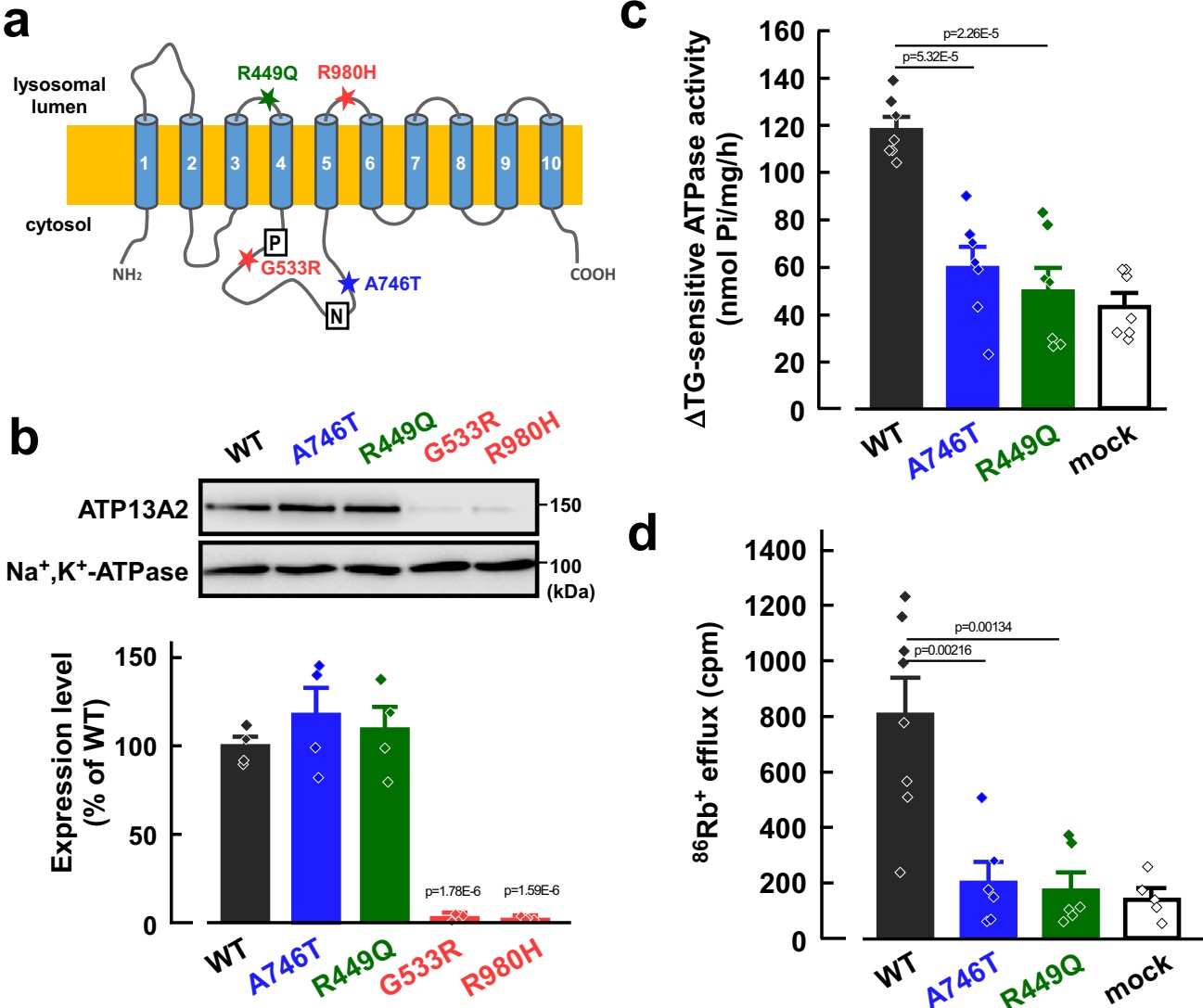

**Fig. 2 | Effects of PD-associated mutations on ATP13A2-derived activity.**
**a** Topology model of human ATP13A2. PD-associated mutations (R449Q, G533R, A746T, and R980H), phosphorylation domain (P), and nucleotide-binding domain (N) are indicated. **b** Western blotting for ATP13A2 (~150 kDa) and Na$^+$,K$^+$-ATPase (~100 kDa) in the ATP13A2 WT- and mutants-transfected HEK293 cells. Expression level of ATP13A2 was normalized to that of Na$^+$,K$^+$-ATPase. The expression level of ATP13A2 in WT-transfected cells was taken as 100%. ($n = 4$ independent replicates). Statistical significance was determined by two-tailed unpaired Student's $t$ test.

**c** ΔTG-sensitive ATP13A2 activities were measured in the membrane fractions of the ATP13A2 WT-, A746T, and R449Q-transfected HEK293 cells. ($n = 7$ independent replicates). Statistical significance was determined by two-tailed unpaired Student's $t$ test. **d** The $^{86}$Rb$^+$ effluxes in the ATP13A2 WT-, A746T- and R449Q-transfected cells. ($n = 5–8$ independent replicates). Statistical significance was determined by two-tailed unpaired Student's $t$ test. All data are presented as mean ± SEM. Source data are provided as a source data file.

(Fig. 4b). The ATP13A2-dependent $^{86}$Rb$^+$ efflux from the lysosomal lumen which is estimated by ATP13A2-knockdown apparently required the presence of ATP (Fig. 4c). In addition, SCH28080 and vonoprazan significantly induced lysosomal alkalinization in SH-SY5Y cells (Fig. 4d). These results suggest that ATP13A2 functions as H$^+$,K$^+$-ATPase in the lysosomes of the cells.

Gastric H$^+$,K$^+$-ATPase consists of two subunits, a catalytic α-subunit and an auxiliary glycosylated β-subunit. These subunits did not express in SH-SY5Y cells (Fig. 4a). To examine whether ATP13A2 (~150 kDa) forms a complex with any β-subunit as well as gastric H$^+$,K$^+$-ATPase, we performed size-exclusion chromatography using the membrane fractions of the exogenously ATP13A2-expressing HEK293 cells and the endogenously ATP13A2-expressing SH-SY5Y cells. Exogenous and endogenous ATP13A2 were detected at the same fractions, which correspond to a lower molecular size than clathrin heavy chain (~200 kDa) (Supplementary Fig. 5), suggesting that ATP13A2 may not

form hetero-multimers with other subunits. These results are consistent with previous reports obtained from cryo-EM structural studies[16,17].

Dysfunction of ATP13A2 impairs degradation of lysosomal substrates, lysosomal-mediated clearance of autophagosomes, and lysosomal exocytosis in dopaminergic neurons, leading to α-synuclein aggregation[7,24,25,35,36]. More than 90% of aggregated α-synuclein in Lewy bodies of PD patients is phosphorylated at residue serine 129 (pS129)[37]. Here, we investigated the effects of P-CABs and bafilomycin A1 on the phosphorylated α-synuclein level in SH-SY5Y cells using an anti-phosphorylated α-synuclein (pS129) antibody (pSyn#64)[38–41]. First, the cells were transiently transfected with an α-synuclein expression vector and recombinant α-synuclein fibrils[42]. As shown in Fig. 4e, the accumulation of phosphorylated α-synuclein was significantly increased by SCH28080, vonoprazan, and bafilomycin A1. In addition, Western blotting using an anti-pan-α-synuclein antibody (MJFR1)

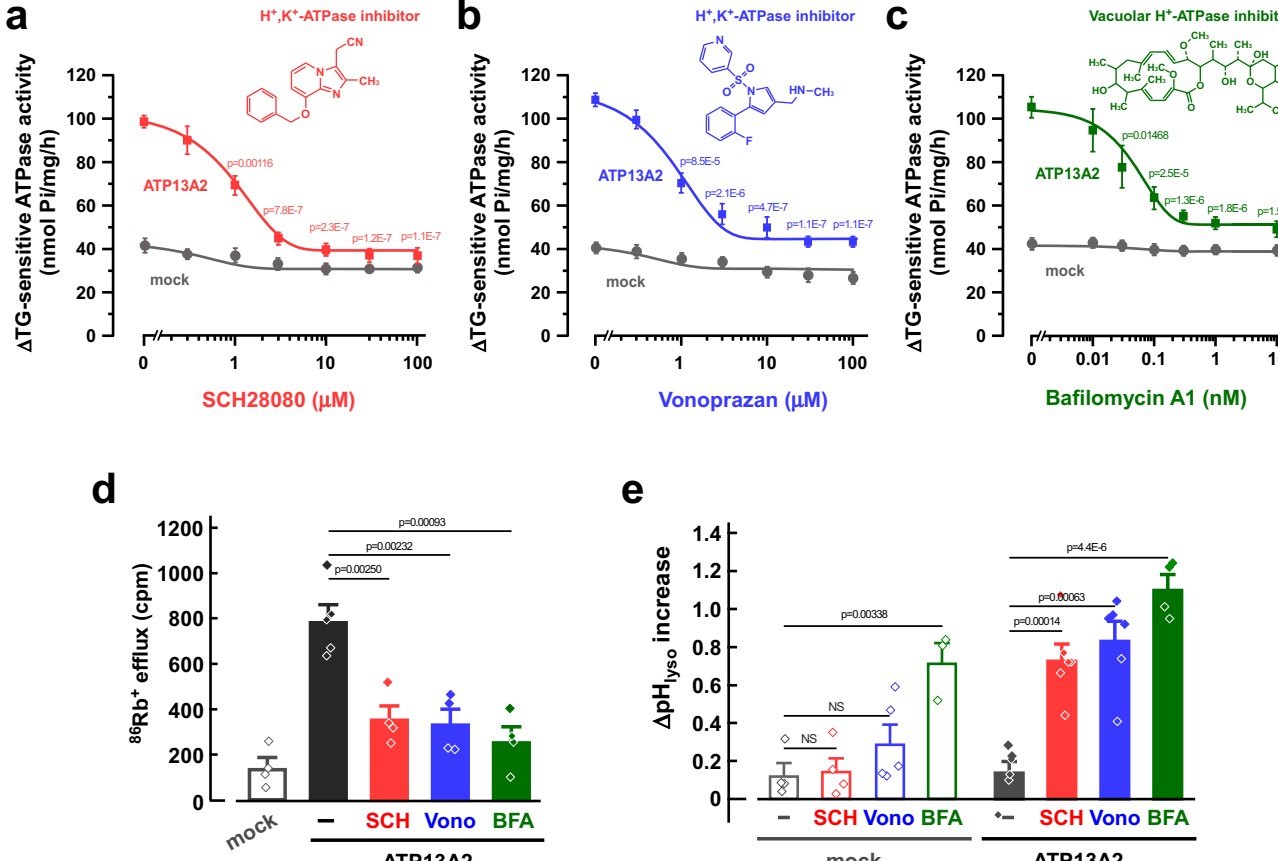

**Fig. 3 | Effects of inhibitors of gastric H⁺,K⁺-ATPase (P-CABs) and vacuolar H⁺-ATPase on exogenous ATP13A2 activity in HEK293 cells. a–c** Concentration-dependent effects of SCH28080 (**a**), vonoprazan (**b**), and bafilomycin A1 (**c**) on the ΔTG-sensitive ATPase activity in the ATP13A2- and mock-transfected HEK293 cells. ($n = 3$–6 independent replicates). Statistical significance was determined by one-way ANOVA. **d** Effects of SCH28080 (SCH; 10 µM), vonoprazan (Vono; 10 µM), bafilomycin (BFA; 5 nM) on the ATP-dependent $^{86}Rb^+$ efflux from lysosomes of the ATP13A2-transfected HEK293 cells. ($n = 4$–5 independent replicates). Statistical significance was determined by two-tailed unpaired Student's $t$ test. **e** Effects of SCH28080 (SCH; 50 µM), vonoprazan (Vono; 50 µM), and bafilomycin A1 (BFA; 5 nM) on the lysosomal pH. In the ATP13A2- and mock-transfected HEK293 cells, the lysosomal pH was measured at 0 and 20 min after treatment of each drug. Then, the pH difference ($\Delta pH_{lyso}$) between 0 and 20 min was calculated. ($n = 3$–6 independent replicates). Statistical significance was determined by two-tailed unpaired Student's $t$ test. All data are presented as mean ± SEM. Source data are provided as a source data file.

showed a significant increase in total α-synuclein level by SCH28080, vonoprazan, and bafilomycin A1 (Fig. 4f). Next, the SH-SY5Y cells stably expressing α-synuclein were established (Fig. 5a). In the cells, SCH28080, vonoprazan, and bafilomycin A1 markedly accelerated the phosphorylated α-synuclein accumulation (Fig. 5b). Finally, SH-SY5Y cells were differentiated to a neuron-like phenotype using retinoic acid and brain-derived neurotrophic factor (BDNF) as previously reported[43] (Fig. 5c). The endogenous accumulation of phosphorylated α-synuclein was significantly increased by SCH28080, vonoprazan, and bafilomycin A1 in the differentiated cells (Fig. 5d). Thus, P-CABs-induced inhibition of H⁺/K⁺-transport by ATP13A2 causes lysosomal alkalinization and α-synuclein accumulation (Fig. 5e).

## Discussion

Our study revealed that ATP13A2 has H⁺/K⁺-transporting activity in lysosomes and that gastric H⁺,K⁺-ATPase inhibitors (P-CABs), a SERCA Ca²⁺-ATPase inhibitor (thapsigargin), and a V-type H⁺-ATPase inhibitor (bafilomycin A1) block the ATP13A2 activity. The P-CABs caused in the lysosomal alkalinization and α-synuclein accumulation in human neuroblastoma cells. In addition, the PD-associated mutants of ATP13A2 (A746T and R449Q) significantly reduced the ATP13A2 activity. Thus, the H⁺/K⁺-transport function of ATP13A2 regulates lysosomal homeostasis and its inhibition may contribute to the pathogenesis of PD (Fig. 5e).

Like other P-type ATPase families, ATP13A2 has several functional domains such as the nucleotide domain, phosphorylating domain, and actuator domain in the large intracellular loop between transmembrane TM4 and TM5[44]. Autosomal recessive mutations near the functional domains of ATP13A2 have been identified in PD patients[3–5]. Here, we examined the effects of four mutations on the function and expression of ATP13A2; G533R and A746T mutations are located between TM4 and TM5, the R449Q mutation is near the TM4, and the R980H mutation is near TM5. Interestingly, the R449Q and A746T mutations impaired the K⁺-transporting activity of the pump without affecting their expression, while the G533R and R980H mutations reduced ATP13A2 expression (Fig. 2). A decrease in ATP13A2 expression by mutation may be due to impaired protein stability and proteasomal degradation[45]. Inhibition of the K⁺-transporting activity of ATP13A2 by P-CABs caused lysosomal dysfunction and phosphorylation of α-synuclein (Figs. 4 and 5). These results suggest that the loss of function of ATP13A2 caused by mutations leads to the pathogenesis of PD.

In the present study, we found that ATP13A2 transports K⁺ from the lysosomal lumen to the cytoplasm and that the activity was inhibited by P-CABs (Figs. 1e, 3a, b). ATP13A2 activity was increased at intraluminal acidic pH of the lysosomes (Fig. 1g) and P-CABs leads to lysosomal alkalinization of the ATP13A2-expressing cells (Fig. 3e), suggesting that ATP13A2 transports H⁺ from the cytoplasm to the

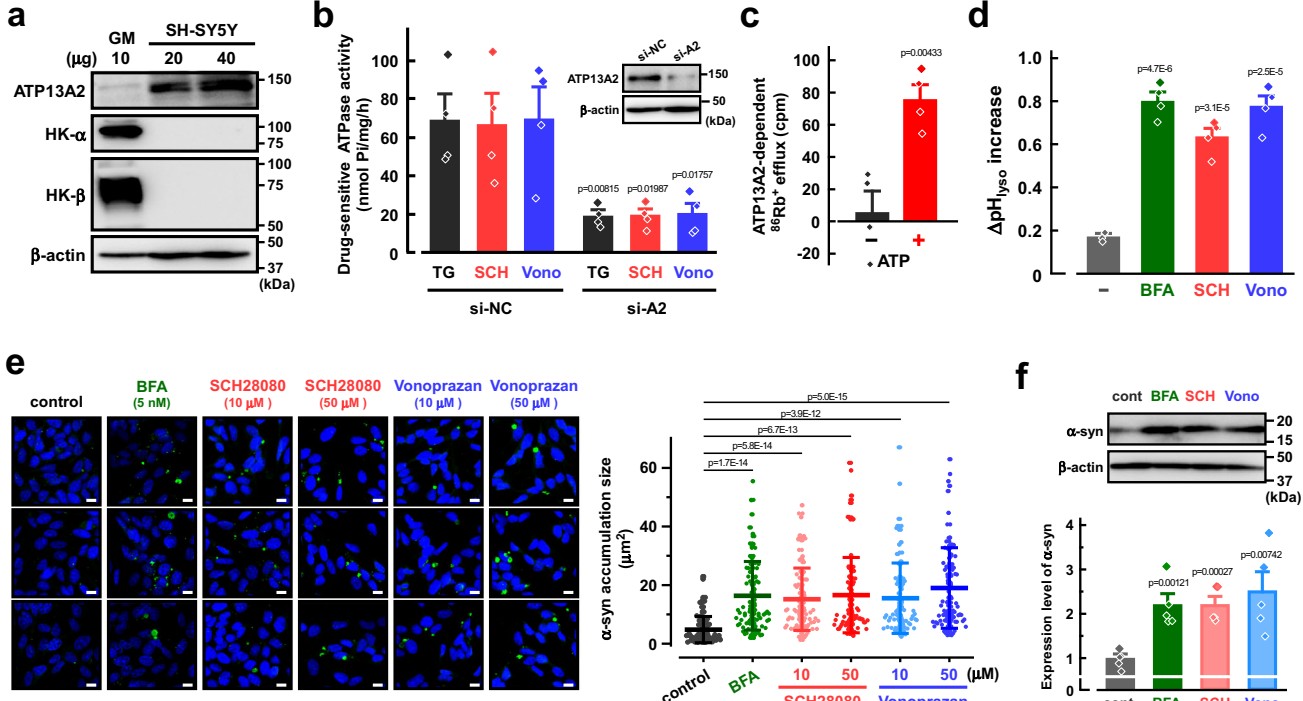

**Fig. 4 | Effects of inhibitors of gastric H⁺,K⁺-ATPase (P-CABs) and vacuolar H⁺-ATPase on endogenous ATP13A2 activity in SH-SY5Y cells. a** Expression of ATP13A2, H⁺,K⁺-ATPase α- and β-subunits (HK-α and HK-β), and β-actin in SH-SY5Y cells and human gastric mucosa (GM). Typical Western blot images in three independent experiments are shown. **b** Effects of TG (5 μM), SCH28080 (SCH; 10 μM), and vonoprazan (Vono; 10 μM) on the ATPase activity in the ATP13A2 siRNA- (si-A2) or negative control siRNA (si-NC)-transfected SH-SY5Y cells. (*n* = 4 independent replicates). Insert shows Western blot images for ATP13A2 and β-actin in the cells. Statistical significance was determined by two-tailed unpaired Student's *t* test. **c** The ATP13A2-dependent ⁸⁶Rb⁺ efflux from lysosomes in the presence and absence of ATP. The ATP13A2-dependent component was assessed as the difference between the ⁸⁶Rb⁺ efflux in the ATP13A2-siRNA- and negative control siRNA-transfected SH-SY5Y cells. (*n* = 4 independent replicates). Statistical significance was determined by two-tailed unpaired Student's *t* test. **d** Effects of bafilomycin A1 (BFA; 5 nM), SCH28080 (SCH; 50 μM), and vonoprazan (Vono; 50 μM) on the lysosomal pH. In SH-SY-5Y cells, the lysosomal pH was measured at 0 and 30 min after treatment of each drug. Then, the pH difference ($\Delta pH_{lyso}$) between 0 and 30 min was calculated. (*n* = 4 independent replicates). Statistical significance was determined by two-tailed unpaired Student's *t* test. **e** Effects of bafilomycin A1 (BFA), SCH28080, and vonoprazan on the phosphorylated α-synuclein (α-syn) level in SH-SY5Y cells. Scale bars, 10 μm. The size of phosphorylated α-synuclein accumulation was measured. (*n* = 88-98 cells in three independent experiments). Statistical significance was determined by two-tailed unpaired Student's *t* test. **f** Western blotting for α-synuclein and β-actin in the SH-SY5Y cells treated with bafilomycin A1 (BFA; 5 nM), SCH28080 (SCH; 10 μM), vonoprazan (Vono; 10 μM). Expression level of α-synuclein was normalized to that of β-actin. The expression level of α-synuclein in SH-SY5Y cells with no drug (cont) was taken as 1. (*n* = 5 independent replicates). Statistical significance was determined by two-tailed unpaired Student's *t* test. Data in (**a**–**d**) and (**f**) are presented as mean ± SEM. Data in e is presented as mean ± SD. Source data are provided as a source data file.

lysosomal lumen as a counter-cation of K⁺. Our phosphorylation assay using [γ-³²P]ATP suggests that the E1 conformation carries H⁺ into the lysosomal lumen and the E2 conformation transports K⁺ to the cytoplasm in the catalytic cycle of ATP13A2 (Supplementary Fig. 4). Tillinghast et al. suggested from their cryo-EM structure that ATP13A2 carries H⁺ in the E1 → E2P transition[17]. These findings suggest that the enzymatic cycle of ATP13A2 is basically similar to that of gastric H⁺,K⁺-ATPase. In the cryo-EM structure of gastric H⁺,K⁺-ATPase, P-CABs bind to a luminal-facing conduit that extends to the K⁺-binding site in the E2P state of H⁺,K⁺-ATPase[46,47]. P-CABs may bind to the E2P state of ATP13A2 as well as with the gastric H⁺,K⁺-ATPase. In addition, we found that nanomolar concentrations of bafilomycin A1 inhibited the ATP13A2 activity (Fig. 3). On the other hand, several dozen μM of bafilomycin A1 inhibit the enzyme activities of other P-type ATPases such as Na⁺,K⁺-ATPase and SERCA Ca²⁺-ATPase[48]. Given that bafilomycin A1 binds to the c subunits of the V₀ domain and inhibits the H⁺ translocation of V-type H⁺-ATPase[49], it may block the H⁺ transport of ATP13A2.

ATP13A2 functions as a polyamine exporter in lysosomes and regulates the cellular polyamine content[12,15,50]. Cryo-EM structures of ATP13A2 revealed the polyamine-binding sites and proposed the polyamine transport mechanism[16–19]. In the E2P state of ATP13A2, a long tunnel (channel-like cavity) is created between TM1-TM2 and TM4-TM5 segments. The tunnel exhibits a negative electrostatic

potential by multiple aspartic and glutamic acid residues. Polyamine is directly recognized by these residues during the initial polyamine-binding steps. However, Sim et al. showed that the cavity in the E2P-Pi state is too narrow (less than ~2 Å) for the polyamine molecule to readily pass and is blocked on the cytosol side[16]. Tomita et al. also suggest that polyamine is stuck at the central kink in the M4 helix and the polyamine-permeating tunnel is sealed on the cytoplasmic side[19]. Although these reports suggest that the binding of the N-terminus of ATP13A2 to membrane lipids may facilitate the polyamine release process, the polyamine transport mechanism has not yet been fully elucidated. Interestingly, K⁺-dependent ATPase activity and ⁸⁶Rb⁺ efflux of ATP13A2 were significantly increased by spermidine (Fig. 1j, k). Thus, the K⁺ transport of ATP13A2 is functionally associated with polyamine transport and may regulate cellular and lysosomal polyamine levels in neurons.

The expression and function of several K⁺ channels have been discovered in lysosomes. Big-conductance Ca²⁺-activated K⁺ (BK) channels (SLO1) are functionally expressed in lysosomes[51,52]. The loss of BK channels leads to abnormal lysosomal storage[51,52]. Transmembrane protein 175 (TMEM175) is also a K⁺ channel located in lysosomes, regulating lysosomal membrane potential, pH stability, and lysosomal fusion[53]. Deficiency of TMEM175 causes α-synuclein aggregation[54] and a loss of dopaminergic neurons[55]. A TMEM175 loss-of-function variant is nominally associated with an accelerated rate of cognitive and motor

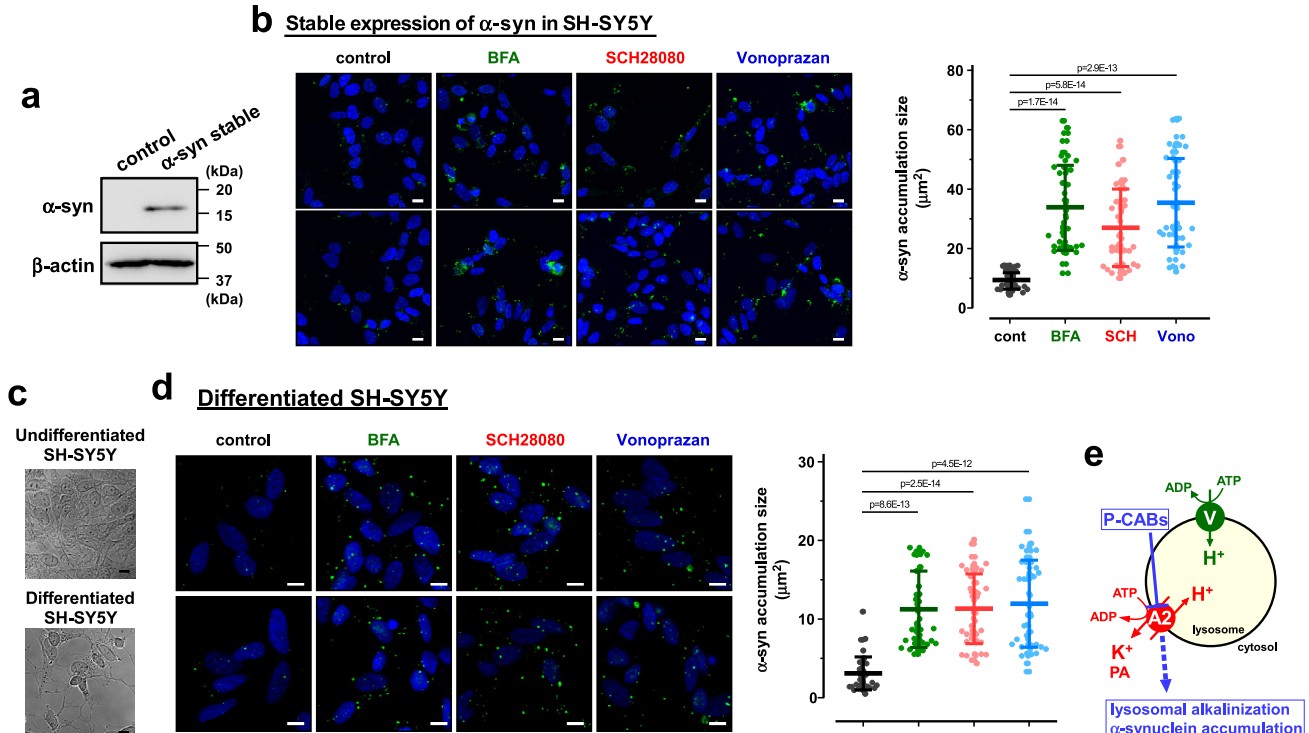

**Fig. 5 | Phosphorylated α-synuclein accumulation in the undifferentiated SH-SY5Y cells stably expressing α-synuclein and differentiated SH-SY5Y cells.**
**a** Expression of α-synuclein (α-syn) and β-actin in the SH-SY5Y cells stably expressing α-synuclein. Typical Western blot images in three independent experiments are shown. **b** Effects of bafilomycin A1 (BFA; 5 nM), SCH28080 (SCH; 10 μM), and vonoprazan (Vono; 10 μM) on the phosphorylated α-synuclein (α-syn) level in the cells stably expressing α-synuclein. Scale bars, 10 μm. The size of phosphorylated α-synuclein accumulation was measured. ($n = 40$ cells in three independent experiments). Statistical significance was determined by two-tailed unpaired Student's $t$ test. **c** Typical microscopic images of differentiated and undifferentiated (control) SH-SY5Y cells in three independent experiments. The control cells were differentiated by treatment with 10 μM retinoic acid for 3 days and 50 ng/ml of BDNF for another 3 days. Scale bars, 10 μm. **d** Effects of bafilomycin A1 (BFA; 5 nM), SCH28080 (SCH; 10 μM), and vonoprazan (Vono; 10 μM) on the phosphorylated α-synuclein (α-syn) level in the differentiated SH-SY5Y cells. Scale bars, 10 μm. The size of phosphorylated α-synuclein accumulation was measured. ($n = 40$ cells in three independent experiments). Statistical significance was determined by two-tailed unpaired Student's $t$ test. **e** Schematic model showing the property of ATP13A2 in the lysosome. ATP13A2 (A2) functions as an H⁺,K⁺-ATPase, and its inhibition by P-CABs leads to lysosomal alkalinization and α-synuclein accumulation. V, vacuolar H⁺-ATPase. PA, polyamines. All data are presented as mean ± SD. Source data are provided as a source data file.

decline with PD[55]. These findings indicate the importance of K⁺-transporting functions in lysosomal homeostasis. Although K⁺ concentration in the lysosomal lumen is regulated to be much lower than that in cytoplasm[26], a K⁺-transporting ATPase has not been reported. Here, we found that inhibition of the K⁺-transporting ATPase function of ATP13A2 by P-CABs causes lysosomal alkalinization and α-synuclein accumulation (Figs. 4, 5). Actually, significant K⁺ efflux from the lysosomal lumen induced by ATP13A2 was observed in the presence of ATP (Fig. 4c). These findings suggest that ATP13A2 can regulate the K⁺ gradient across the lysosomal membrane through the catalytic cycle of ATP hydrolysis.

In conclusion, we revealed that ATP13A2 functions not only as a polyamine transporter but also as H⁺,K⁺-ATPase in lysosomes and regulates lysosomal pH. Our findings resolve a long-standing question about the cation-transporting properties of ATP13A2. The dysfunction of the H⁺/K⁺-transport of ATP13A2 may explain one of the pathogenic mechanisms of PD.

## Methods
### Materials
Spermidine, ouabain, SCH28080, FITC-dextran, retinoic acid, brain-derived neurotrophic factor (BDNF), and anti-β-actin antibody (AC-74) were obtained from Sigma-Aldrich. Thapsigargin was from Cayman Chemical. Anti-Xpress-tag antibody and lipofectamine 3000 were from Thermo Fisher Scientific. Anti-Na⁺,K⁺-ATPase α1-isoform antibody (C464.6) was from Santa Cruz Technology. Anti-phosphorylated α-

synuclein antibody (pSyn#64), bafilomycin A1, omeprazole, valinomycin, and G-418 were from Fujifilm Wako. Anti-ATP13A2, anti-Lamp2, anti-calnexin, and anti-clathrin heavy chain antibodies were from Cell Signaling Technology. Anti-H⁺,K⁺-ATPase α-subunit (1H9) and β-subunit (2B6) antibodies were from Medical & Biological Laboratories. Horse-radish peroxidase-conjugated anti-mouse and rabbit IgGs were from Millipore. Anti-pan-α-synuclein antibody (MJFR1) was from Abcam. Alexa Fluor 568- and 488-conjugated IgG antibodies were from Abcam. 4′,6-Diamidino-2-phenylindole (DAPI) was from Dojindo Laboratories. Saponin was from Nakarai Chemicals. Fetal bovine serum (FBS) was from Nichirei bioscience. Western Lightning ECL Pro and [γ-³²P] ATP were from PerkinElmer. ⁸⁶RbCl was from Polatom. All other reagents were of molecular biology grade or the highest grade of purity available.

### Plasmid construction
Full-length cDNA encoding human ATP13A2 was amplified by PCR using KOD-Plus DNA polymerase (Toyobo) and the following primers (sense primer: 5′-atagaattcatgagcgcgaga-3′, and antisense primer: 5′-aattctagactacctcaggg-3′). The PCR product was ligated into pcDNA4/His C vector equipped with Xpress-tag by using EcoRI and XbaI restriction sites. Site-directed mutagenesis for preparing the D513N, A746T, R449Q, G533R, and R980H mutants was performed using the KOD-Plus-mutagenesis Kit (Toyobo). The cDNA sequences of wild-type (WT) and mutants were verified using Big Dye Terminator V3.1 Cycle Sequencing Kit (Life Technologies) and an ABI PRISM 3500 sequencer (Applied Biosystems).

## Cell culture and transfection

Human embryonic kidney HEK293 cells (kindly provided by Prof. Shinji Asano of Ritsumeikan University[33]) were maintained in DMEM (Fujifilm Wako) supplemented with 100 units/ml penicillin, 100 μg/ml streptomycin, and 10% FBS. The cells were transfected with the vector by using PEI-Max (Polysciences) and cultured for 24 h. ATP13A2 protein having Xpress-tag at its N-terminus was expressed. The ATP13A2-knockout HEK293 cells were created using the Guide-it CRISPR/Cas9 system (Takara Biotechnology). The sgRNA target sequence for human ATP13A2 (5′- CACCGGTCAGGGTCCCATAACCGGT-3′, Nippon Gene) was cloned into the pGuide-it-ZsGreen1 Vector (Clontech), then transfected to HEK293 cells using lipofectamine 3000. Transfected cells were sorted by green fluorescence with a FACSAria SORP cytometer (BD Biosciences).

Human neuroblastoma SH-SY5Y cells (EC94030304-F0; DS Pharma Biomedical.) were maintained in Ham's F12/DMEM (Fujifilm Wako) supplemented with 100 units/ml penicillin, 100 μg/ml streptomycin and 10% FBS. The cells were transfected with the ATP13A2 siRNA (5′-gcaucuucauccucuaccgtt-3′ (sense) and 5′-cgguagaggaugaagaugctt-3′ (antisense), Nippon Gene) by using lipofectamine 3000 and cultured for 48 h. For the transient expression of α-synuclein, SH-SY5Y cells were transfected with pCMV-SNCA (α-synuclein expression vector) and recombinant α-synuclein fibrils by MultiFectam and cultured for 4 days according to the manufacturer's protocols of α-synuclein aggregation assay kit (Cosmo Bio). For stable expression of α-synuclein, SH-SY5Y cells were transfected with the pCMV-SNCA (α-synuclein expression vector; Cosmo Bio) using lipofectamine 3000 and cultured for 24 h. The transfected cells were selected in the presence of 500 μg/ml G-418.

## Differentiation of SH-SY5Y cells

SH-SY5Y cells were differentiated as described previously[43]. SH-SY5Y cells were treated with 10 μM retinoic acid in Ham's F12/DMEM containing 2% FBS for 3 days, followed by replenishing with fresh medium containing 10 μM retinoic acid in 0.5% FBS for another 3 days. Then, the cells were treated with DMEM/F12 containing 50 ng/ml of BDNF and 0.5% FBS for 3 days.

## Preparation of membrane fractions

HEK293 and SH-SY5Y cells cultured in 10-cm dishes were washed with PBS and incubated in low ionic salt buffer ($0.5$ mM $MgCl_2$ and 10 mM Tris-HCl, pH 7.4) supplemented with the protease inhibitor cocktail (10 μg/ml aprotinin, 10 μg/ml phenylmethylsulfonyl fluoride, 1 μg/ml leupeptin and 1 μg/ml pepstatin A) at 4 °C for 10 min. Subsequently, they were homogenized with 40 strokes in a Dounce homogenizer and diluted with an equal volume of a solution containing 500 mM sucrose and 10 mM Tris-HCl (pH 7.4). The homogenized suspension was centrifuged at $800 \times g$ for 10 min. The supernatant was centrifuged at $100,000 \times g$ for 90 min, and the pellet was suspended in a solution containing 250 mM sucrose and 5 mM Tris-HCl (pH 7.4).

## Western blotting

The membrane fractions of HEK293 and SH-SY5Y cells were solubilized in lysis buffer supplemented with 1.5% N-dodecyl b-D-maltoside (DDM) and 0.3% cholesterol hemisuccinate (CHS) were mixed with electrophoresis buffer (50 mM Tris-HCl (pH 6.8), 10% glycerol, 100 mM dithiothreitol, 2% SDS, and bromphenol blue) and separated by electrophoresis on a 9% or 15% SDS-polyacrylamide gel. The gels were then transferred to the PVDF membrane. The membranes were blocked with the solution containing 5% skim milk and 0.1% Tween 20 in TBS (150 mM NaCl and 25 mM Tris-HCl, pH 7.4) for 1 h at room temperature. For primary antibodies, Anti-Xpress-tag (1:5,000), $Na^+,K^+$-ATPase α1-isoform (1:3,000), ATP13A2 (1:1,000), $H^+,K^+$-ATPase α- and β-subunits (1:5,000), pan-α-synuclein (MJFR1; 1:2,000), clathrin heavy chain (1:2,000), and β-actin (1:5,000) antibodies were used. For

secondary antibodies, HRP-conjugated anti-mouse or rabbit IgG antibody was used at 1:3,000 dilution. The signals were visualized using Western Lightning ECL Pro, SuperSignal West Femto Maximum Sensitivity Chemiluminescent Substrate, and a LAS-4000 system (FujiFilm).

## Immunocytochemistry

HEK293 cells cultured on collagen type1-coated coverglass (Iwaki) were fixed with ice-cold methanol for 5 min and washed with PBS containing $0.1$ mM $CaCl_2$ and 1 mM $MgCl_2$ (PBS++). The cells were permeabilized with permeabilization buffer containing 0.3% Triton X-100 and 0.1% bovine serum albumin (BSA) in PBS + + for 15 min at room temperature. Non-specific binding of the antibody was blocked with a solution containing 20 mM phosphate buffer (pH 7.4), 450 mM NaCl, 16.7 % goat serum, and 0.3% Triton X-100, and the cells were incubated with the anti-Xpress-tag (1:100), anti-LAMP2 (1:100), and anti-calnexin (1:100) antibodies overnight at 4 °C. The cells were washed with the permeabilization buffer and incubated with the Alexa Fluor 568-conjugated anti-mouse IgG antibody (1:100) and Alexa Fluor 488-conjugated anti-rabbit IgG antibody (1:100) for 1 h at room temperature. The cells were washed with PBS++ and incubated with DAPI (1:1,000) for visualizing DNA. Immunofluorescence images were visualized by using a Zeiss LSM 780 laser scanning confocal microscope.

## Measurement of ATP-hydrolyzing activity

The ATP-hydrolyzing activity was measured as described previously[56]. The activity of membrane fractions (50 μg of protein) was mainly measured in a solution containing 4 mM NaCl, 160 mM KCl, 2 mM $MgSO_4$, 1 mM ATP, 1 mM $NaN_3$, 17 mM HEPES-Tris (pH 6.5, 7.0, and 7.4). To prepare the solutions at pH 5.5 and 6.0, HEPES was replaced with MES. After incubation for 30 min at 37 °C, the reaction was terminated by the addition of the ice-cold stop solution containing 12% perchloric acid and 3.6% ammonium molybdate, and the released inorganic phosphate was measured. $Na^+$ and $K^+$ were replaced with N-methyl-D-glucamine (NMDG).

## Measurement of $^{86}Rb^+$ efflux

HEK293 cells ($7 \times 10^4$ cells) were transfected with the expression vectors and cultured for 24 h in the medium containing $^{86}RbCl$ ($2.5 \times 10^5$ cpm/ml). For permeabilization of the plasma membrane, the cells were incubated for 5 min with a solution containing 120 mM RbCl, 2 mM $MgCl_2$, 1 mM $CaCl_2$, and PIPES-NaOH (pH 7.2), and 20 μg/ml saponin, and washed with a wash buffer containing with 150 mM RbCl and PIPES-NaOH (pH 7.2). The permeabilized cells were then incubated for 15 min with an efflux buffer containing 150 mM RbCl, 30 mM NaCl, 2 mM $MgCl_2$, 1 mM $CaCl_2$, 2 mM ATP, 100 μM ouabain, and PIPES-NaOH (pH 7.2). The incubation was stopped by cooling on ice and the plates were washed twice with an ice-cold wash buffer, and the radioactivity was measured by liquid scintillation. Differences in the radioactivity in the presence and absence of ATP were evaluated as an ATP-dependent $^{86}Rb^+$ efflux.

## FITC-dextran-based measurement of lysosomal pH

SH-SY5Y cells and HEK293 cells were cultured with a medium containing 50 μg/ml FITC-dextran for 48 h. The cells were harvested with trypsin-EDTA and washed with PBS. The fluorescence of FITC-dextran in the cells was measured using a FACScant II flow cytometer (BD Biosciences). The voltages of forward scatter (FSC) and side scatter (SSC) were set at 250 and 420 mV, respectively. The fluorescence intensity of FITC in cell groups with FSC and SSC greater than 50,000 was measured. Lysosomal pH was calculated from the fluorescence of FITC-dextran with the standard curve obtained from cells resuspended in the solution containing monensin (100 μM) at pH (4.5, 5.0, 5.5, 6.0, 6.5, 7.0, and 7.5).

## Size-exclusion chromatography

The membrane fractions of ATP13A2-transfected HEK293 cells and SH-SY5Y cells were solubilized in lysis buffer supplemented with 1.5% DDM and 0.3% CHS. Insoluble material was removed by ultracentrifugation at $366,000 \times g$, 30 min at 4 °C. The samples were then injected into a Superose 6 10/300 GL column (GE Life Sciences) connected to the AKTAexplorer 10 XT FPLC system (GE Healthcare) at 4 °C, equilibrated in a running buffer containing 50 mM Tris pH 7.5, 150 mM NaCl, 0.03% DDM, and 0.0015% CHS. Fractions of 1 mL were collected, precipitated with 10% trichloroacetic acid (TCA), and analyzed by Western blotting.

## Phosphorylation assay

Membrane fractions (50 µg) were added to the reaction buffer (2 mM $MgCl_2$, 5 mM $NaN_3$, 17 mM HEPES, pH 6.5 or 7.4) with or without 20 mM KCl. The reaction was initiated on ice by adding [γ-$^{32}$P] ATP (2 µCi) and stopped after 20 s with the stop solution (20% trichloroacetic acid and 10 mM phosphoric acid). After precipitation on ice for 20 min, samples were centrifuged ($20,000 \times g$, 10 min, 4 °C). The pellet was washed with ice-cold stop solution and distilled water and dissolved in sample buffer comprising 2% SDS, 2.5% dithiothreitol, 10% glycerol, and 50 mM Tris-HCl, pH 6.8, and subjected to the 5% SDS-polyacrylamide gel under acidic conditions at pH 6.0[57]. The radioactivity was visualized and quantified by digital autoradiography of the dried gel using Typhoon FLA 9500.

## Detection of phosphorylated α-synuclein in SH-SY5Y cells

The undifferentiated SH-SY5Y cells transiently and stably expressing α-synuclein and the differentiated SH-SY5Y cells by retinoic acid and BDNF were treated with SCH28080, vonoprazan, and bafilomycin A1 for 48 h. The cells were fixed with ice-cold methanol for 5 min, washed with Tris-buffered saline (TBS), and permeabilized with permeabilization buffer containing 0.3% Triton X-100 and 0.1% BSA in TBS for 15 min at room temperature. Non-specific binding of the antibody was blocked with 2% BSA in TBS, and the cells were incubated with an anti-phosphorylated α-synuclein antibody (pSyn#64) (1:100) in Can Get Signal immunostain (Toyobo) for 15 h at 4 °C. The cells were washed with TBS and incubated with the Alexa Fluor 488-conjugated anti-mouse IgG antibody (1:100) for 1 h at room temperature. Immunofluorescence images were visualized by using a Zeiss LSM 700 laser scanning confocal microscope. The area (dimension) of the fluorescent signal due to phosphorylated α-synuclein was measured by Zen3.3 software (Zeiss).

## Human tissue procurement

Human normal gastric mucosa was obtained from the dissected stomach of a Japanese gastric cancer patient (Female, 67 years) in accordance with the recommendations of the Declaration of Helsinki and with ethics committee approval of the University of Toyama (No. R2017085). Informed consent was obtained from the patient at Toyama University Hospital.

## Statistical analysis

Results are shown as mean ± SEM. Differences between groups were analyzed by one-way analysis of variance, and correction for multiple comparisons was made by using Tukey's multiple comparison test. A comparison between the two groups was made by using two-tailed unpaired Student's $t$ test. Statistically significant differences were assumed at $P < 0.05$.

## Reporting summary

Further information on research design is available in the Nature Portfolio Reporting Summary linked to this article.

## Data availability

Source data are available as a Source Data file. Source data are provided with this paper.

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

## Acknowledgements

This work was supported by Grants-in-Aid for Scientific Research (KAKENHI) from Japan Society for the Promotion of Science (JSPS) (to H.S. (22H02801), Ta.F. (20K07258), S.N. (21H03365), and T.S. (22K06827)), by JSPS Core-to-Core Program, B. Asia-Africa Science Platforms, Tamura Science & Technology Foundation, and Platform for drug discovery, informatics, and structural life science.

## Author contributions

Ta.F. and H.S. designed all experiments and wrote the paper. Ta.F., S.N., P.W., S.Z., A.Y., T.S., Y.T., and T.O. performed the experiments. Ts.F. and H.T. gave conceptual advice. All authors reviewed the results and approved the final version of the manuscript.

## Competing interests

The authors declare no competing interests.
