## [Peer Review File · Nature Communications]

Parkinson's disease-associated ATP13A2/PARK9 functions as a lysosomal H⁺,K⁺-ATPaseReviewers' Comments:

Reviewer #1:

Remarks to the Author:

The manuscript from Fujii et al. studied the functions of Parkinson's disease-associated ATP13A2/PARK9 as a novel H⁺, K⁺-ATPase in lysosomes. They analyzed the properties of cation transport (Mn²⁺, Fe²⁺, Zn²⁺, and Ca²⁺) in different experimental conditions. They assessed distinct pharmacological assays, two different cellular models, and D523N, A746T, R449Q, G533R, and R980H ATP13A2 mutants.

The study is interesting, clearly presented, and the experiments are well-performed. However, the choice of the cellular models should be improved and reinforced with a more suitable one.

Supp.Fig.1A: The scale bar value is not clearly indicated. No quantification between ATP13A2 and LAMP1 colocalization is shown. LAMP2 is a better lysosomal-related marker, as we observe a poor colocalization between LAMP1 and ATP13A2.

Supp.Fig.1B: The authors used different cell lines. In the D513N-transfected cells, there is both the expression of ATP13A2 WT- and -D513N. The authors should knock out ATP13A2 in HEK293 cells before transfect ATP13A2 D513N to reveal the contribution of the mutant ATP13A2 D513N.

Figure 4: The assessment of α -synuclein aggregation in SH-SY5Y cells is relatively weak. Before claiming an effect, a panel of methodologies has to be used, such as biochemical assays (immunoblot with specific or conformational antibodies, filter retardation assay, α -synuclein aggregation Kit HTRF assay.....). As said below, SH-SY5Y cells are not a perfect model to answer this question. What would be the effects in the context of an overexpressing α -synuclein cell line?

Figure 4: The authors should replicate some salient results in a cellular model closer to the dopaminergic cell line than undifferentiated SH-SY5Y cells. For example, the authors may consider differentiated SH-SY5Y cells or primary culture of mouse dopaminergic neurons, or iPSC-derived dopaminergic neurons.

Minor comments: please correct lysosomal alkalization by lysosomal alkalinization

Reviewer #2:

Remarks to the Author:

Points to address:

- 1). The results are indeed novel and noteworthy.
- 2). The significance is wide-ranging.
- 3). However, the authors claim that ATP13A2 is a PD-related, lysosomal ATPase. The evidence presented does not necessarily support this conclusion (for various reasons, as stated below, but mainly due to the variety of the consequences of well-known inhibitors to other ATP-dependent transporters, which are known to work on different types of ATPases, not just P-type ATPases?
- 4). Flaws, some questionable interpretation of data presented?
- 5). The methodology, except for the use and consequences of the use of inhibitors, seems sound.
- 6). Reproducibility: There is enough information to reproduce their findings.

Major comments to the authors:

The gastric H,K-ATPase activity is stimulated by valinomycin. Did the authors catch an effect by valinomycin?

Is it known whether this transporter transports alpha-synuclein (as it does with other polycations, such as spermidine)?

A major concern is the characterization of the transporter. Is there an effect of ouabain, on transport activity, or could there be interference from Na,K-ATPase activity (outside of the changes in Na⁺, or K⁺)?

In some cases, inhibitors such as TG, can inhibit the activity of P-type ATPases, but others, such as bafilomycin A1, is an inhibitor of vacuolar ATPases (mentioned below). Thus, the inhibitors target 2 different classes of ATPase activities? Is there an explanation for this inhibition?

The immunofluorescent image may indicate (Suppl. Fig. 1) that ATP13A2 might be an ER-resident transporter (or other organelles)? Is there co-localization with PDI or calnexin for the ER or other markers for other organelles? Did the authors have quantitative co-localization data, like a Pearson correlation coefficient, for co-localization?

Is there an effect of omeprazole? While it is not a P-CAB, it is a good inhibitor (and any member of this family of drugs) of the gastric H,K-ATPase?

The effect of bafilomycin A1 is curious, because this inhibitor, unlike the others, does not inhibit P-type ATPases, but rather, vacuolar ATPases?

Figure 1: This ATPase shows virtually no activity at physiological pH's? If this is correct, what is the function of this ATPase? Is it activated only when the cytosolic pH's drop?

Can the transport of polyamines also induce alkalinization of lysosomal lumina?

Throughout: for clarification, "intraluminal" or "cytoplasmic" should be indicated, especially for the transport of H⁺ and K⁺.

Mutations resulting in decreased expression are not commented upon? Is there anything known about these mutations, and the level of expression, particularly of ATP13A2?

What are the polyamines that cannot be transported when ATP13A2 is mutated, and how are these related to defects seen in PD?

Is the gradient created/maintained of K⁺ across the lysosomal membrane and cytosol powered by the ATPase sufficiently powered by a cycle of ATP hydrolysis?

In vivo, is it known whether ATP13A2 is a heterodimer (like the gastric H,K-ATPase)?

Minor comments:

p. 3, line 40: hallmark, change to: hallmarks

p. 3, line 51: blacked, change to: blocked

p. 4, line 73: change, change to: changes

Abstract: TG as an inhibitor of the ATPase should also be mentioned; this finding is novel and potentially impactful.

p. 7, line 128: transports, change to: transport

p. 8, line 159: those, change to: their

p. 9, line 174: add "not" after "has."

p. 9, line 177: add "a" before "luminal-facing"

p. 9, line 179: add "with the" before "gastric"

p. 9, line 181: transports, change to: transport

p. 9, line 182: leaded, change to: lead

p. 9, line 186: are, change to: have been

p. 9, line 190: Does the inhibition of TMEM 175 result in accumulation in lysosomes of α -synuclein? Is it known in PD whether alpha-synuclein is stored in lysosomes or is secreted from damaged brain tissue?

p. 9, line 194: "a" before K+

p. 9, line 187: Does ATP13A2 carry H⁺ from the E1-P + ADP to the E2-P.K conformation all the way, or does it transport it across the lysosomal membrane in exchange for K⁺? Please clarify, as a comparison to the proposed catalytic cycle of the gastric H,K-ATPase?

RESPONSE LETTER

We thank all reviewers for their careful evaluation and constructive comments on the revision of our manuscript. We have performed a variety of additional experiments and modified the manuscript following all the comments made by the reviewers. We provide a marked version of the manuscript where all alterations are highlighted in red color in the marked manuscript.

Response to Reviewer #1

The manuscript from Fujii et al. studied the functions of Parkinson's disease-associated ATP13A2/PARK9 as a novel H⁺, K⁺-ATPase in lysosomes. They analyzed the properties of cation transport (Mn²⁺, Fe²⁺, Zn²⁺, and Ca²⁺) in different experimental conditions. They assessed distinct pharmacological assays, two different cellular models, and D523N, A746T, R449Q, G533R, and R980H ATP13A2 mutants.

The study is interesting, clearly presented, and the experiments are well-performed. However, the choice of the cellular models should be improved and reinforced with a more suitable one.

Response: We appreciate your insightful and helpful comments and suggestions. We performed additional experiments and modified the manuscript following all your comments. We have addressed all issues point by point, as mentioned below. We provide a marked version of the manuscript where all alterations are highlighted in red color in the marked manuscript.

Supp.Fig.1A: The scale bar value is not clearly indicated. No quantification between ATP13A2 and LAMP1 colocalization is shown. LAMP2 is a better lysosomal-related marker, as we observe a poor colocalization between LAMP1 and ATP13A2.

Response: Following your and another reviewer's suggestions, immunocytochemical experiments were performed using anti-LAMP2 (a lysosomal marker) and anti-calnexin (an ER marker) antibodies. We calculated the colocalization coefficient of ATP13A2/LAMP2 and ATP13A2/calnexin by the Pearson correlation coefficient analysis. In the cells, ATP13A2 largely overlapped with LAMP2 (colocalization coefficient; 0.73 ± 0.03) but not calnexin (colocalization coefficient; 0.13 ± 0.05) (please see Supplementary Fig. 1). The scale bar values are indicated in the legend. Please see the Results section (on page 5, lines 69-71) and the Methods section (on page 15, lines 287 and 288 and page 18, line 369).

Supp.Fig.1B: The authors used different cell lines. In the D513N-transfected cells, there is both the expression of ATP13A2 WT- and -D513N. The authors should knock out ATP13A2 in HEK293 cells before transfect ATP13A2 D513N to reveal the contribution of the mutant ATP13A2 D513N.

Response: Following your suggestion, we constructed the ATP13A2-knockout HEK293 cells using the Guide-it CRISPR/Cas9 system with the sgRNA target sequence for human ATP13A2 (5'- CACCGGTCAGGGTCCCATAACCGGT-3'). In the KO cells, the faint expression of endogenous ATP13A2 disappeared and the basal Δ TG-sensitive ATPase activity was decreased (Please see new Supplementary Fig. 2a,b). Then, the ATP13A2 D513N or the ATP 13A2 WT was transfected to the KO cells. In the D513N-transfected KO cells, no ATP13A2-derived ATPase activity was

detected, while the WT-transfected KO cells showed the significant activity (please see new Supplementary Fig. 2b). Please see the Results section (on page 5, lines 81-86) and the Methods section (on page 16, lines 312-317).

Figure 4: The assessment of α -synuclein aggregation in SH-SY5Y cells is relatively weak. Before claiming an effect, a panel of methodologies has to be used, such as biochemical assays (immunoblot with specific or conformational antibodies, filter retardation assay, α -synuclein aggregation Kit HTRF assay.....). As said below, SH-SY5Y cells are not a perfect model to answer this question. What would be the effects in the context of an overexpressing α -synuclein cell line?

Response: Thank you very much for the suggestion. In Figure 4e, we used α -synuclein aggregation Kit; the SH-SY5Y cells were transiently transfected with α -synuclein expression vector and recombinant α -synuclein fibrils. Following your suggestion, we performed Western blotting using anti- α -synuclein antibody. Expression level of α -synuclein protein was significantly increased by the treatments with SCH28080, vonoprazan, and bafilomycin A1. In addition, we established the SH-SY5Y cells stably expressing α -synuclein (please see new Fig. 4f). In the cells, SCH28080, vonoprazan, and bafilomycin A1 also accelerated α -synuclein accumulation (new Fig. 5a,b). Please see the Results section (on page 10, lines 194-200) and the Methods section (on pages 16 and 17, lines 322-328, page 18, line 356, and page 21, lines 426-428).

Figure 4: The authors should replicate some salient results in a cellular model closer to the dopaminergic cell line than undifferentiated SH-SY5Y cells. For example, the authors may consider differentiated SH-SY5Y cells or primary culture of mouse dopaminergic neurons, or iPSC-derived dopaminergic neurons.

Response: Thank you very much for the suggestion. According to the method reported previously (Guan et al. *Experimental Neurology*, 2017), we established the differentiated SH-SY5Y cells by using retinoic acid and BDNF. The differentiated cells displayed the neuron-like phenotype with elongation and branched neurites morphology as previously reported (please see new Fig. 5c). In the differentiated cells, the accumulation of endogenous α -synuclein was significantly increased by SCH28080, vonoprazan, and bafilomycin A1 (please see new Fig. 5d). Please see the Results section (on page 10, lines 200-204) and the Methods section (on page 17, lines 330-335 and page 21, lines 426-428).

Minor comments: please correct lysosomal alkalization by lysosomal alkalinization

Response: Thank you for pointing out the mistake. We corrected lysosomal alkalization by lysosomal alkalinization.

Response to Reviewer #2

Points to address:

- 1). The results are indeed novel and noteworthy.
- 2). The significance is wide-ranging.
- 3). However, the authors claim that ATP13A2 is a PD-related, lysosomal ATPase. The evidence presented does not

necessarily support this conclusion (for various reasons, as stated below, but mainly due to the variety of the consequences of well-known inhibitors to other ATP-dependent transporters, which are known to work on different types of ATPases, not just P-type ATPases?)

4). Flaws, some questionable interpretation of data presented?

5). The methodology, except for the use and consequences of the use of inhibitors, seems sound.

6). Reproducibility: There is enough information to reproduce their findings.

Response: We appreciate your valuable comments and insightful suggestions. We performed additional experiments and modified the manuscript following all your comments. We have addressed all issues point by point, as mentioned below. We provide a marked version of the manuscript where all alterations are highlighted in red color in the marked manuscript.

Major comments to the authors:

The gastric H,K-ATPase activity is stimulated by valinomycin. Did the authors catch an effect by valinomycin?

Response: Following your suggestion, we examined the effect of valinomycin on the ATP13A2-derived ATPase activity. Valinomycin had no significant effect on the activity. Please see the Results section (on page 8, lines 141-143).

Is it known whether this transporter transports alpha-synuclein (as it does with other polycations, such as spermidine)?

Response: Thank you very much for the comment. Dysfunction of ATP13A2 prevents polyamine export from the lysosomal lumen, leading to lysosomal polyamine accumulation, intraluminal alkalinization of lysosomes, and aggregation of α -synuclein in lysosomes. It has been reported that ATP13A2 plays a protective role against aggregation of α -synuclein by maintaining the integrity of the lysosomal membrane. It also promotes the ATPase-independent secretion of α -synuclein through nanovesicles (Si et al., Int. J. Mol. Sci., 2021). These findings suggest that ATP13A2 may not directly transport α -synuclein across the lysosomal membrane.

A major concern is the characterization of the transporter. Is there an effect of ouabain, on transport activity, or could there be interference from Na,K-ATPase activity (outside of the changes in Na⁺, or K⁺)?

Response: Following your suggestion, we examined the effects of ouabain on the ATP13A2-derived ATPase activity and ⁸⁶Rb⁺ efflux in the ATP13A2-transfected HEK293 cells. Ouabain (30 μ M) had no significant effect on the ATPase activity and ⁸⁶Rb⁺ efflux (please see new Supplementary Figure 3a,b). These results suggest that the plasma membrane of the cells is completely permeabilized and the effect of Na⁺,K⁺-ATPase is eliminated. Please see the Results section (on page 6, lines 98-102).

In some cases, inhibitors such as TG, can inhibit the activity of P-type ATPases, but others, such as bafilomycin A1, is an inhibitor of vacuolar ATPases (mentioned below). Thus, the inhibitors target 2 different classes of ATPase activities? Is there an explanation for this inhibition?

The effect of bafilomycin A1 is curious, because this inhibitor, unlike the others, does not inhibit P-type ATPases, but rather, vacuolar ATPases?

Response: As you pointed out, the inhibitory effect of bafilomycin A1 on the ATP13A2-derived activity is comparable to that on vacuolar ATPase activity; nanomolar concentrations of bafilomycin A1 significantly inhibited the ATP13A2 activity. On the other hand, several dozen μM of bafilomycin A1 have been reported to inhibit the enzyme activities of other P-type ATPases such as Na^+, K^+ -ATPase and SERCA Ca^{2+} -ATPase (Bowman et al., PNAS, 1988). These findings suggest that bafilomycin A1 can target two different classes of ATPase activity. Given that bafilomycin A1 binds to the c subunits of V_0 domain and inhibits the H^+ translocation of V-type H^+ -ATPase (Wang et al., Nat. Commun., 2021), it may block the H^+ transport of ATP13A2. Structural analysis using cryo-electron microscopy may be necessary to elucidate the detailed molecular mechanism by which bafilomycin A1 inhibits ATP13A2. Please see the Discussion sections (on page 12, lines 240-244).

The immunofluorescent image may indicate (Suppl. Fig. 1) that ATP13A2 might be an ER-resident transporter (or other organelles)? Is there co-localization with PDI or calnexin for the ER or other markers for other organelles? Did the authors have quantitative co-localization data, like a Pearson correlation coefficient, for co-localization?

Response: Following your and another reviewer's suggestions, immunocytochemical experiments were performed using anti-LAMP2 (a lysosomal marker) and anti-calnexin (an ER marker) antibodies. We calculated the colocalization coefficient of ATP13A2/LAMP2 and ATP13A2/calnexin by the Pearson correlation coefficient analysis. In the cells, ATP13A2 largely overlapped with LAMP2 (colocalization coefficient; 0.73 ± 0.03) but not calnexin (colocalization coefficient; 0.13 ± 0.05) (please see Supplementary Fig. 1). Please see the Results section (on page 5, lines 69-71) and the Methods section (on page 15, lines 287 and 288 and page 18, line 369).

Is there an effect of omeprazole? While it is not a P-CAB, it is a good inhibitor (and any member of this family of drugs) of the gastric H^+, K^+ -ATPase?

Response: Following your suggestion, we examined the effects of omeprazole on the ATP13A2-derived ATPase activity. Unlike P-CABs, neither omeprazole nor acid-treated omeprazole ($30 \mu\text{M}$) inhibited the ATP13A2-derived ATPase activity (new Supplementary Fig. 3c). These results suggest that molecular structure of the K^+ -binding site (P-CAB-binding site, but not omeprazole-binding site) of ATP13A2 is similar to that of gastric H^+, K^+ -ATPase. Please see the Results section (on pages 7 and 8, lines 137-141).

Figure 1: This ATPase shows virtually no activity at physiological pH's? If this is correct, what is the function of this ATPase? Is it activated only when the cytosolic pH's drop?

Response: Thank you very much for the suggestion. In Fig. 1g, we showed that acidic pH (below pH 7) increased the ATP13A2-derived ATPase activity but not at pH 7.4. In this experiment, we used the membrane fractions prepared from the ATP13A2-expressing HEK293 cells. In these samples, both intraluminal and cytoplasmic sides of ATP13A2 are exposed to the outside solutions at acidic or neutral pH. On the other hand, in the $^{86}\text{Rb}^+$ efflux assay (please see new Fig. 1e-f), the cells were treated with saponin to permeabilize the plasma membrane. In this condition, lysosome is intact. Therefore, cytosolic side but not intraluminal side of lysosome faces on the outside solution. In fact, the $^{86}\text{Rb}^+$ efflux was detected under cytosolic (outside) solution at pH 7.4, suggesting that the intraluminal

acidic pH of the lysosomes facilitates the K⁺-transport of ATP13A2. Please see the Results section (on page 6, lines 96-110).

Can the transport of polyamines also induce alkalinization of lysosomal lumina?

Response: Thank you for the question. ATP13A2 transports polyamines from the lysosomal lumen to the cytoplasm (van Veen et al., Nature, 2020). They measured the lysosomal pH using FITC-dextran and showed no significant change in the lysosomal pH by spermine exposure under normal conditions. However, exposure of spermine induces abnormal alkalinization of the lysosomal lumen in the ATP13A2-deficient cells (van Veen et al., Nature, 2020). These findings suggest that the lysosomal accumulation of polyamine causes to alkalinization of the lysosomal lumen.

Throughout: for clarification, “intraluminal” or “cytoplasmic” should be indicated, especially for the transport of H⁺ and K⁺.

Response: We are sorry for the unclear description. We clarified the direction of H⁺ and K⁺ transport “intraluminal” or “cytoplasmic”.

Mutations resulting in decreased expression are not commented upon? Is there anything known about these mutations, and the level of expression, particularly of ATP13A2?

Response: Thank you very much for the suggestion. It has been reported that decrease in ATP13A2 expression by mutation is due to impaired protein stability and proteasomal degradation (Podhajska et al., Plos One, 2012). We added this report in the Discussion section (on page 11, lines 222-224).

What are the polyamines that cannot be transported when ATP13A2 is mutated, and how are these related to defects seen in PD?

Response: Thank you very much for the questions. Polyamines are polycationic and aliphatic molecules participating in numerous cellular processes, including gene transcription and translation, protein synthesis, cellular proliferation, and differentiation, but they become toxic at high concentrations. Abnormal cytoplasmic polyamine level promotes the aggregation and fibrillization of α -synuclein in PD (Antony et al., J. Biol. Chem., 2003). We added this description in the Introduction section (on page 3, lines 46-52).

Is the gradient created/maintained of K⁺ across the lysosomal membrane and cytosol powered by the ATPase sufficiently powered by a cycle of ATP hydrolysis?

Response: Thank you very much for the suggestion. So far, no K⁺-transporting ATPase in lysosomes has been reported. In new Figure 4c using SH-SY5Y cells, the ATP13A2-dependent ⁸⁶Rb⁺ efflux from the lysosomal lumen which is estimated by ATP13A2-knockdown apparently required the presence of ATP. In addition, the inhibition of the K⁺-transporting ATPase function of ATP13A2 by P-CABs causes lysosomal alkalinization in SH-SY5Y cells (please see new Fig. 4d). These findings suggest that ATP13A2 can regulate the K⁺ gradient across the lysosomal membrane

through the catalytic cycle of ATP hydrolysis. Please also see the Results section (on page 9, lines 178-181) and the Discussion section (on page 13, lines 268-274).

In vivo, is it known whether ATP13A2 is a heterodimer (like the gastric H,K-ATPase)?

Response: Following your suggestion, size-exclusion chromatography using Superose 6 10/300 GL column was performed with the solubilized membrane fractions of the exogenously ATP13A2-expressing HEK293 cells and the endogenously ATP13A2-expressing SH-SY5Y cells (please see new Supplementary Figure 5). Both exogenous and endogenous ATP13A2 (~150 kDa) were detected at the same fractions with elution volumes of 14-17 ml. On the other hand, clathrin heavy chain (~200 kDa) was detected at the fractions (11-13 ml). In this analysis, the molecular size of proteins in the 11-13 ml fractions is higher than in the 14-17 ml fractions. It is noted that these eluted fractions of ATP13A2 are similar to that of purified monomeric ATP13A2 in the size-exclusion chromatography on Superose 6 10/300 GL column previously reported in cryo-EM structural studies (Sim et al., Mol. Cell, 2021; Tillinghast et al., Mol. Cell, 2021). The systems genomics approach shows that most of the genes and pathways in PD pathogenesis are intact in the SH-SY5Y cells (Krishna et al., BMC Genomics, 2014). These findings suggest that ATP13A2 may not form hetero-multimers with other subunits in neurons. Please also see the Results section (on pages 9 and 10, lines 182-190) and Methods section (on page 15, line 288, page 18, line 356, and page 20, lines 405-412).

Minor comments:

p. 3, line 40: hallmark, change to: hallmarks

p. 3, line 51: blacked, change to: blocked

p. 4, line 73: change, change to: changes

p. 7, line 128: transports, change to: transport

p. 8, line 159: those, change to: their

p. 9, line 174: add "not" after "has."

p. 9, line 177: add "a" before "luminal-facing"

p. 9, line 179: add "with the" before "gastric"

p. 9, line 181: transports, change to: transport

p. 9, line 182: leaded, change to: lead

p. 9, line 186: are, change to: have been

p. 9, line 194: "a" before K⁺

Response: Thank you very much for your careful reading of our manuscript. We apologize for our mistakes. We corrected these following your suggestion.

Abstract: TG as an inhibitor of the ATPase should also be mentioned; this finding is novel and potentially impactful.

Response: Following your comment, we described the inhibition of ATP13A2 by TG in the revised Abstract (Please see page 2, lines 25-27)

p. 9, line 190: Does the inhibition of TMEM 175 result in accumulation in lysosomes of α -synuclein? Is it known in PD whether alpha-synuclein is stored in lysosomes or is secreted from damaged brain tissue?

Response: Thank you very much for the comment. TMEM175 regulates lysosomal membrane potential, pH stability, and lysosomal fusion (Cang et al., Cell, 2015). A deficiency of TMEM175 causes α -synuclein aggregation (Jinn et al., PNAS, 2017) and a loss of dopaminergic neurons (Wie et al., Nature, 2021). A TMEM175 loss-of-function variant reported in PD patients is nominally associated with an accelerated rate of cognitive and motor decline with PD (Wie et al., Nature, 2021). We added these reports in the Discussion section (on page 13, lines 264-266). As you pointed out, α -synuclein is accumulated in lysosomes of neurons by the lysosomal storage disorder, causing neuronal cell death (Srinivasan et al., Front. Med., 2021). Recently, it has also been reported that the aggregated α -synuclein is secreted into the extracellular space via SNARE-dependent lysosomal exocytosis in the brain of a mouse model of synucleinopathy (Xie et al., Nat. Commun., 2022).

p. 9, line 187: Does ATP13A2 carry H⁺ from the E1-P + ADP to the E2-P.K conformation all the way, or does it transport it across the lysosomal membrane in exchange for K⁺? Please clarify, as a comparison to the proposed catalytic cycle of the gastric H,K-ATPase?

Response: Following your suggestion, we measured the phosphorylation level of ATP13A2 using [γ -³²P]ATP. In the membrane fractions of ATP13A2-transfected HEK293 cells, a phosphorylated band of ATP13A2 (~150 kDa) was detected at pH 6.5 in the absence of KCl (please see new Supplementary Fig. 4a). This band was not observed in the mock-transfected cells (new Supplementary Fig. 4a). The ATP13A2-derived phosphorylated band was significantly decreased by the addition of KCl (20 mM) (please new Supplementary Fig. 4a). In the absence of KCl, the phosphorylated level of ATP13A2 at pH 6.5 was higher than that at pH 7.4 (new Supplementary Fig. 4b). These results are similar to the case for gastric H⁺,K⁺-ATPase (Asano et al., J. Biochem., 2000), suggesting that ATP13A2 carries H⁺ from the (H⁺)E1P to the (K⁺)E2P in its catalytic cycle across the lysosomal membrane. Please also see the Results section (on pages 8 and 9, lines 157-167), the Discussion section (on page 12, lines 233-237), and Methods section (on page 20, lines 414-423).

Reviewers' Comments:

Reviewer #1:

Remarks to the Author:

The authors have addressed most of my concerns. However, I remained unconvinced by the α -synuclein (α -syn) aggregation results in SH-SY-5Y cells (Fig. 4 Panels). The immunocytochemistry experiment is not precise about what is recognized and quantified by the antibody labelling. Moreover, the antibody reference is not indicated. Western-blot experiment shows only the monomeric form of synuclein, which is not related to α -synuclein (α -syn) aggregation. This experiment does not answer the question of α -synuclein aggregation readout. Dedicated experiments are necessary to claim α -synuclein (α -syn) aggregation in suitable experimental models.

RESPONSE LETTER

We thank the editor and reviewers for their careful evaluation and constructive comments on our revised manuscript. We modified the manuscript following the comments made by the editor and reviewer #1. We also corrected the manuscript and figures according to the Author Checklist. We provide a marked version of the manuscript where all alterations are highlighted in red color in the marked manuscript.

Response to Editor

With regards to the referee's outstanding concerns for the data presented in Figure 4, we suggest it either be removed, or the results toned down to only claim what the data clearly shows, with caveats to the limitations of the results plainly stated to the reader.

Response: We appreciate your kind consideration. Following your suggestion, we replaced the description of "α-synuclein aggregation" with "α-synuclein accumulation" or "phosphorylated α-synuclein accumulation" in Figure 4 and related parts in the manuscript. Please see the Results section (on page 9, line 169, and page 10, lines 194-209), the Discussion section (on page 11, line 228, and page 13, line 274), and Figures 4e, 5b, and 5d and their legends.

As Reviewer #1 pointed out, clarification of the α-synuclein aggregation (ubiquitination) in the cells is an important issue. However, all anti-α-synuclein antibodies we tested for this study do not have strong interactions with the aggregated form in Western blotting. We would be very obliged if Reviewer #1 understands the limitation of the present study.

In the immunocytochemistry experiment (Figure 4e), we detected the phosphorylation of α-synuclein at residue serine 129 (pS129) using an anti-phosphorylated α-synuclein antibody as previously reported (Nonaka et al., J. Biol. Chem. 2010). In this report by Nonaka et al., they suggested that phosphorylation (Ser129) of α-synuclein is associated with the α-synuclein aggregation. More than 90% of the aggregated α-synuclein in Lewy bodies of Parkinson's disease patients is phosphorylated at residue serine 129 (Fujiwara et al., Nat. Cell Biol., 2002). The likelihood, the accumulation of phosphorylated α-synuclein (pS129) is associated with the α-synuclein aggregation and pathogenesis of Parkinson's disease.

Response to Reviewer #1

The authors have addressed most of my concerns. However, I remained unconvinced by the α-synuclein (α-syn) aggregation results in SH-SY-5Y cells (Fig. 4 Panels). The immunocytochemistry experiment is not precise about what is recognized and quantified by the antibody labelling. Moreover, the antibody reference is not indicated. Western-blot experiment shows only the monomeric form of synuclein, which is not related to α-synuclein

(α -syn) aggregation. This experiment does not answer the question of α -synuclein aggregation readout. Dedicated experiments are necessary to claim α -synuclein (α -syn) aggregation in suitable experimental models.

Response: Thank you very much for your valuable comments. We completely agree with your suggestion. As you pointed out, it was very difficult to show the α -synuclein aggregation in our experimental condition since we used the soluble fraction of the cells for Western blotting in Figure 4f. When the insoluble fraction was used in Western blotting, we found weak signals which likely indicate aggregated α -synuclein. Most of the aggregated α -synuclein may be contained in the insoluble fraction. However, all anti- α -synuclein antibodies we tested for this study do not have strong interactions with the aggregated form in Western blotting. We would be very obliged if you understand the limitation of the present study.

In the immunocytochemistry of Figure 4e, we used an anti-phosphorylated α -synuclein antibody (clone: pSyn#64) which reacts specifically to α -synuclein phosphorylated at residue serine 129 (pS129). This antibody has widely been used for detecting phosphorylated α -synuclein in human cell lines including SH-SY5Y cells, mouse brain, and Lewy Bodies in the human brain (Saito et al., *J. Neuropathol. Exp. Neurol.*, 2003; Lee et al., *J. Cell Sci.*, 2013; Delic et al., *J. Comp. Neurol.*, 2018; Terada et al., *J. Biol. Chem.*, 2018). Regarding the quantification method in Figure 4e, the area (dimension) of the fluorescent signal of phosphorylated α -synuclein was measured by Zen3.3 software. Therefore, it is fair to present the qualifications as the degree of phosphorylated α -synuclein accumulation.

Based on these reasons, we replaced the description of " α -synuclein aggregation" with " α -synuclein accumulation" or "phosphorylated α -synuclein accumulation" in Figure 4 and related parts in the manuscript. Please see the Results section (on page 9, line 169, and page 10, lines 194-209), the Discussion section (on page 11, lines 228, and page 13, line 274), the Methods section (on page 20, lines 430-442), and Figures 4e, 5b, and 5d and their legends.